# A STRUCTURED MATRIX METHOD FOR NONEQUISPACED NEURAL OPERATORS

## ABSTRACT

The computational efficiency of many neural operators, widely used for learning solutions of PDEs, relies on the fast Fourier transform (FFT) for performing spectral computations. However, as FFT is limited to equispaced (rectangular) grids, this limits the efficiency of such neural operators when applied to problems where the input and output functions need to be processed on general non-equispaced point distributions. We address this issue by proposing a novel method that leverages batch matrix multiplications to efficiently construct Vandermonde-structured matrices and compute forward and inverse transforms, on arbitrarily distributed points. An efficient implementation of such *structured matrix methods* is coupled with existing neural operator models to allow the processing of data on arbitrary non-equispaced distributions of points. This work effectively implements the formulation of the Fourier transform within the domain of any given problem–an aspect that has long been mentioned in literature but has never been realized until now. With extensive empirical evaluation, we demonstrate that the proposed method allows one to extend neural operators to very general point distributions with significant gains in training speed over baselines, while retaining or improving accuracy.

## 1 INTRODUCTION

Partial Differential Equations (PDEs) are extensively used to mathematically model interesting phenomena in science and engineering (Evans, 2010). As explicit solution formulas for PDEs are not available, traditional numerical methods such as finite difference, finite element, and spectral methods (Quarteroni & Valli, 1994) are extensively used to simulate PDEs. Despite their tremendous success, the prohibitively high computational cost of these methods makes them infeasible for a variety of contexts in PDEs ranging from high-dimensional problems to the so-called *many query* scenarios (Karniadakis et al., 2021). This high computational cost also provides the rationale for the development of alternative *data driven* methods for the fast and accurate simulation of PDEs. Hence, a wide variety of machine learning algorithms have been proposed recently in this context. These include physics-informed neural networks (PINNs) (Raissi et al., 2019), MLPs, and CNNs for simulating parametric PDEs (Zhu & Zabaras, 2018; Lye et al., 2020; 2021; Wandel et al., 2020) as well as graph-based algorithms (Sanchez-Gonzalez et al., 2020; Pfaff et al., 2020; Brandstetter et al., 2022; Equer et al., 2023), to name a few.

However, as solutions of PDEs are expressed in terms of the so-called *solution operators*, which map input functions (initial and boundary data, coefficients, source terms) to the PDE solution, *Operator learning*, i.e., learning the underlying operators from data, has emerged as a dominant framework for applying machine learning to PDEs. Existing operator learning algorithms include, but are not limited to, operator networks (Chen & Chen, 1995), DeepONets (Lu et al., 2019; Mao et al., 2020; Cai et al., 2021), attention-based methods such as (Kissas et al., 2022; Cao, 2021; Prasthofer et al., 2022), and neural operators (Kovachki et al., 2021a; Li et al., 2020c;d; Raonić et al., 2023).

Within this large class of operator learning algorithms, Neural Operators based on *non-local* spectral transformations, such as the Fourier Neural operator (FNO) (Li et al., 2020a) and its variants (Li et al., 2021; Pathak et al., 2022) have gained much traction and are widely applied. Apart from favorable theoretical approximation properties (Kovachki et al., 2021b; Lanthaler et al., 2023), FNOs are attractive due to their expressivity, simplicity, and computational efficiency. We provide a brief

review of the formulation of the FNO in **SM** A.1. A key element in the computational efficiency of FNO and its variants lies in the fact that its underlying convolution operation is efficiently carried out in Fourier space with the *fast Fourier transform* (FFT) algorithm. It is well-known that FFT is only (log-)linear in computational complexity with respect to the number of points at which the underlying input functions are sampled. However, this computational efficiency comes at a cost as the recursive structure of FFT limits its applications to inputs sampled on the so-called *Regular* or *equispaced Cartesian (Rectangular) grids*, see **SM** Figure 5 left for an illustration. This is a major limitation in practice. In real-world applications, where information on the input and output signals is measured by sensors, it is not always possible to place sensors only on an equispaced grid. Similarly, when data is obtained through numerical simulations, often it is essential to discretize PDEs on irregular grids, such as those adapted to be refined to capture relevant spatially localized features of the underlying PDE solution or on unstructured grids that fit the complex geometry of the underlying domain. See **SM** Figure 5 for examples of such non-equispaced distributions of sample points or *point clouds*.

Several methods have been proposed in the literature to address this limitation of FNOs and modify/enhance it to handle data on non-equispaced points. A straightforward fix would be to interpolate data from non-equispaced point distributions to equispaced grids. However, as shown in (Li et al., 2022), the resulting procedure can be computationally expensive and/or inaccurate. Consequently, (Li et al., 2022) proposes a geometry-aware FNO (Geo-FNO) that appends a neural network to the FNO to learn a deformation from the physical space to a regular grid. Then, the standard FFT can be applied to the latent space of equispaced grid points. This learned diffeomorphism corresponds to an adaptive moving mesh (Huang & Russell, 2010). Factorized-FNO (F-FNO) builds upon the Geo-FNO, introducing an additional bias term in the Fourier layer and performing the Fourier transform over each dimension separately (Tran et al., 2023). The non-equispaced Fourier PDE solver (NFS) uses a vision mixer (Dosovitskiy et al., 2020) to interpolate from a non-equispaced signal onto a regular grid, again applying the standard FNO subsequently (Lin et al., 2022). All these methods share the same design principle, i.e., given inputs on non-equispaced points, *interpolate* or transform this data into a regular grid and then apply FNO.

This observation leads to a natural question: *Is there a complementary approach where truncated spectral transformations, such as the discrete Fourier transform (DFT) inside FNO, can be performed efficiently on arbitrary point distributions?*

In this paper, we answer this question affirmatively to design neural operators that can handle inputs and outputs on arbitrary point distributions. More concretely,

- We present a simple, yet powerful, structured matrix method that leverages the *Vandermonde structure* of the matrices corresponding to the truncated spectral transformations, to easily and accurately extend Fourier-based Neural Operators to arbitrary point distributions. A *novel* PyTorch implementation of this structured matrix method is also presented.

- We *replace* the standard truncated spectral transformations that underpin many widely-used neural operators by the proposed *structured matrix method*, to obtain neural operators that can efficiently handle input and output data on arbitrary point distributions on domains with arbitrary geometries.

- We present a suite of extensive numerical experiments to demonstrate that a variety of neural operators, based on truncated spectral transformations computed using the proposed structured matrix method, outperform baselines in terms of both accuracy and efficiency (training speed) in various scenarios involving input/output data on arbitrary point distributions.

## 2 METHODS

In this section, we describe the construction of Vandermonde matrices to compute truncated spectral transformations, such as the DFT and spectral harmonics, which form the foundations of many widely-used neural operators such as FNO (Li et al., 2020a), UFNO (Wen et al., 2022), F-FNO (Tran et al., 2023) and SFNO (Bonev et al., 2023). To this end, we start with a short description of this matrix construction below.

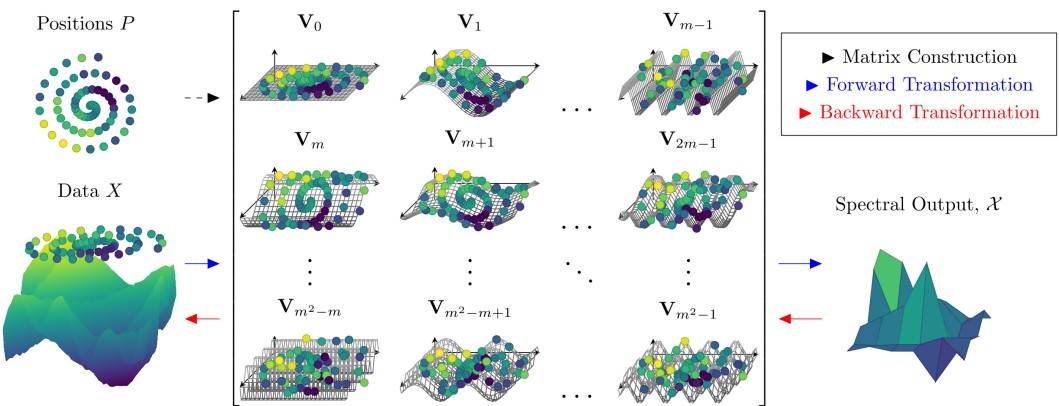

Figure 1: Outline of the proposed approach. Taking the positions of the data points, we build a Vandermonde-structured matrix, which acts as the foundation for the forward and inverse spectral transformations within the layers of the neural operator. The subscript $\mathbf{V_j}$ refers to the $j^{th}$ row of the matrix from Equation (6).

**On Vandermonde Structured Matrices.** A Vandermonde structured matrix can be defined via nodes $x_j \in \mathbb{R}$ or $\mathbb{C}, j \in \{0, 1, \ldots, m-1\}$, in a geometric progression along each column (or row), defined by

$$\mathbf{V}_{j,k} = \left[x_j^k\right]_{j,k=0}^{m-1,n-1}. \tag{1}$$

In the case that the nodes represent the primitive $n^{th}$ roots of unity, this Vandermonde matrix turns into the DFT matrix. The DFT matrix is symmetric, unitary, and periodic, hence leading to an elegant factorization. The resulting factorization can be used to obtain a radix-2 algorithm that can efficiently compute the DFT and its inverse so-called the FFT algorithm with $O(n \log n)$ complexity (Cooley & Tukey, 1965; Yavne, 1968; Johnson & M., 2007).

To motivate an alternative computational realization of DFT, we recall that the Fourier transform is an integral operator. Approximating it with a quadrature rule for discretization, we multiply each point value of the underlying function by the sinusoidal basis function for a given mode and sum the resulting terms. This process is then repeated for each mode. In contrast to the recursive, butterfly FFT algorithm, this interpretation can be generalized to a fast algorithm based on Vandermonde-structured matrices of the form in (1), which in turn, can be efficiently implemented through *batched matrix multiplications*.

**Forward Transformations.** The forward transformation computes the Fourier representation of a given function by using Vandermonde-structured matrices. For 1D data, the corresponding Vandermonde-structured matrix is

$$\mathbf{V}_{j,k} = \frac{1}{\sqrt{n}} \left[e^{-i(jp_k)}\right]_{j,k=0}^{m-1,n-1}. \tag{2}$$

Here, $\boldsymbol{p} = [p_0, p_1, \ldots, p_{n-1}]^T$ is the vector of the positions of data points at which the underlying function is sampled, normalized to range between $0$ and $2\pi$. $n$ is the number of data points, $m$ the number of modes, and $i = \sqrt{-1}$. If these points are equispaced on the unit circle, the output of (2) is exactly equivalent to the DFT, and therefore the corresponding algorithm is identical to the FFT. However, this form is clearly realized even for arbitrary non-equispaced point distributions.

The 2D Fourier transform is equivalent to 1D Fourier transforms along each axis. Therefore, the transformation on any 2D lattice, i.e, the tensor product of 1D point distributions along each axis (see **SM** Figure 5), can be performed by constructing two Vandermonde matrices, $\mathbf{V}_1$ and $\mathbf{V}_2$, corresponding to the positions of data points along each axis. Given $X \in \mathbb{R}^{n \times n}$ or $\mathbb{C}^{n \times n}$ as the data matrix containing values of the underlying function, sampled on the non-equispaced lattice, the Vandermonde-structured matrix can be used as a transformation of this data to the Fourier space $\mathcal{X} \in \mathbb{C}^{m \times m}$ given via

$$\mathcal{X} = \mathbf{V}_1 X \mathbf{V}_2^T. \tag{3}$$

The transforms for non-equispaced 1D and non-equispaced 2D rectangular lattices are already been proposed in the literature (Bagchi & Mitra, 1999), yet generalizations to more general distribution have not been not considered to date.

**Extension to N-dimensional point clouds.** Next, we would like to extend this construction beyond lattices to point clouds, i.e., arbitrary distributions of sampling points. To do this, we store again the positions of the sampling points as $P = [\boldsymbol{p}_0, \boldsymbol{p}_1, \ldots, \boldsymbol{p}_{N-1}] \in \mathbb{R}^{n \times N}$ and normalize the points' values to a range between $[0, 2\pi]$. The exponents of the corresponding Vandermonde matrix are a flattened tensor product of the selected modes in each dimension. This results in the following matrix,

$$\mathbf{V}_{j,k} = \sqrt{\frac{N}{n}} \left[ e^{-i\left(\sum\limits_{l=0}^{N}\left(\lfloor \frac{j}{m^l} \rfloor \mod m\right)P_{k,l}\right)} \right]_{j,k=0}^{m^N-1,n-1} \tag{4}$$

Given two-dimensional point-cloud data $X \in \mathbb{R}^n$ or $\mathbb{C}^n$ as the data matrix containing values of the underlying function, sampled on any arbitrary distribution of points $P \in \mathbb{R}^{n \times 2}$, the Vandermonde-structured matrix can be used as a basis transformation of this data to the Fourier space $\mathcal{X} \in \mathbb{C}^{m^2}$ given via,

$$\mathcal{X} = \mathbf{V}X \tag{5}$$

with matrix $\mathbf{V}$ below,

$$\mathbf{V} = \sqrt{\frac{2}{n}} \begin{bmatrix} e^{-i(0\mathbf{p_0}^T+0\mathbf{p_1}^T)} \\ e^{-i(1\mathbf{p_0}^T+0\mathbf{p_1}^T)} \\ \vdots \\ e^{-i((m-1)\mathbf{p_0}^T+0\mathbf{p_1}^T)} \\ e^{-i(0\mathbf{p_0}^T+1\mathbf{p_1}^T)} \\ e^{-i(1\mathbf{p_0}^T+1\mathbf{p_1}^T)} \\ \vdots \\ e^{-i((m-1)\mathbf{p_0}^T+(m-1)\mathbf{p_1}^T)} \end{bmatrix}. \tag{6}$$

This can be readily implemented in a single shot using tensorized methods, eliminating the need for Python's less efficient loop constructs. Likewise, it leverages methods which have been highly optimized for 3D or 4D tensors, such as *torch.bmm()* and *torch.matmul()* for PyTorch.

**Extension to Spherical Harmonics.** The above method for constructing these structured matrices is not just limited to Fourier transforms and can thus be integrated in other Neural Operators. Simply using the basis functions of other spectral transformations leads to a formulation that is adapted for the underlying spectral transform algorithms, such as wavelets or Laplace transforms. We exemplify such an extension to the spherical harmonics below.

The spherical harmonics are derived as the eigenfunctions of the Laplacian on the sphere. Given the order $l$ and degree $m$, $-l \leq m \leq l$, the associated harmonics can be explicitly calculated and arranged into a Vandermonde matrix as,

$$\mathbf{V}_{j,k} = \left[ Ce^{im\phi_k}\mathcal{P}_l^m(\cos m\theta_k) \right]_{j,k=0}^{m_{max},n}$$
$$m = j - (\lfloor \sqrt{j} \rfloor^2 + \lfloor \sqrt{j} \rfloor), \quad l = \lfloor \sqrt{j} \rfloor, \tag{7}$$

for any point $k = 0, \ldots, n$ with polar angle $\theta_k$ and azimuth $\phi_k$, where $C$ is a normalization constant and $\mathcal{P}_l^m$ is the associated Legendre polynomial. The total number of modes, $m_{max}$, is taken as the square of the order, as all degrees are computed for a given order. Then, the forward transformation for the spherical harmonics is readily obtained from the data $X$ by multiplying as in (5).

**Backward Transformations.** Following the construction presented in this section, the transformation from the spectral domain back to the physical space is immediately calculated by multiplying the data in the frequency domain by the conjugate transpose of the forward transformation matrix, $\bar{\mathbf{V}}^T$. For training on problems with varying distributions, these matrices could be constructed in the

loop for each training sample, significantly slowing down the training time. Instead, we opt for a simple 3D tensor construction, therefore, allowing us to forgo the construction of a separate matrix for the adjoint transformation. Consequently, this trick leads to further improvements in computational efficiency.

**Computational Complexity.** The most notable feature of FFT is its computational efficiency. Calculating the Fourier coefficients of a 1D signal, sampled at $n$ points, by using the brute force DFT, costs $O(n^2)$. In contrast, the FFT algorithm computes these coefficients with $O(n \log n)$ complexity.

Hence, it is natural to wonder why one should reconsider matrix multiplication techniques in our setting. In this context, we observe that the maximum performance gain with FFT occurs when the FFT computes all the Fourier coefficients, or modes, of an underlying signal. Furthermore, peak efficiency is reached for points on a dyadic interval. While the number of modes to compute may be truncated, the interconnected nature of the self-recursive radix-2 FFT algorithm makes it difficult in practice to attain peak efficiency. We refer the reader to **SM** Figure 2 for a visual representation. Therefore, reported performance gains by new FFT algorithms are often optimistic. Thus, in the case of truncated modes, matrix multiplication techniques could be competitive vis a vis computational cost.

Moreover, for neural operators, only a small subset of nonzero modes are required to approximate the operator (Li et al., 2020a). This implies that for 1D problem, the Vandermonde matrix has a fixed number of rows denoted by $m$, while the number of columns grows with the problem size $n$. Therefore, the computational complexity of the proposed transformations by a Vandermonde matrix cost $O(mn)$ as the Vandermonde-structure can be fully determined using $O(mn)$ as opposed to $O(n^2)$ (Gohberg & Olshevsky, 1994b; Pan, 2001; Gohberg & Olshevsky, 1994a), and hence the number of points is independent of the number of modes. A detailed discussion on the computational complexity of the structured matrix method proposed here, both for Fourier transforms and Spherical harmonics is provided in **SM** A.3. We illustrate how the computational complexity of the proposed structured matrix method pans out with respect to that of standard FFT within neural operators, for example within the Fourier Neural Operator. We present an ablation study in **SM** Figure 3, varying the number of modes and observing the computation time. The proposed structured matrix method is clearly more efficient when using 64 or fewer modes. Given that the typical number of truncated modes for FNO ranges between 12 and 20 for most problems (Li et al., 2020a), this figure suggests that structured matrix methods will be faster to run in practice.

## 3 EXPERIMENTAL RESULTS

In this section, our aim is to investigate the performance of the proposed structured matrix method (SMM) within various neural operator architectures on a challenging suite of diverse PDE tasks.

**Implementation, Training Details and Baselines.** A key contribution of this paper is a new implementation of the Vandermonde structured matrix multiplications in *PyTorch*, which enables us to efficiently compute Fourier and Inverse Fourier transforms. Within a neural network, an efficient $O(n)$ algorithm must also be parallelizable to handle batches, as this massively speeds up the training process. Batches of data with the same or different point distributions are easily handled by the *torch.matmul()* and *torch.bmm()* functions.

In all experiments, we use a simple grid search to select the hyperparameters, which nearly always converge to the same values; ADAM optimizer with a learning rate of 0.005, scheduler step 10, gamma decay of 0.97. We train all models until convergence. We also use the L1-loss function, which produced both a lower L1-error and L2-error than the L2-loss. The test error was measured in all experiments as the relative L1 error.

As baselines, we use the geometric diffeomorphism (Geometric Layer), described in (Li et al., 2022) in experiments where the underlying domain has a complicated, non-equispaced geometry, when applicable. For the one dimensional experiment, we use a cubic interpolation scheme on the nonequispaced data. For the experiments on a lattice, we take the model's performance over the original grid as a baseline. To apply the SFNO in the sperical example, we explore RBF interpolation schemes with both a Gaussian kernel with variance 0.1 and a linear kernel.

Table 1: Performance results for all experiments, comparing the SMM approaches to various baselines. The SMM approach offers clear advantages in speeding up training time, improving the testing error, or both across a variety of experiments with both equispaced grids and unusual geometries.

| Model | Method | Training Time (per epoch) | L1 Test Error |
|---|---|---|---|
| **1D: Burgers' Equation** | | | |
| *Equispaced Distribution:* | | | |
| FNO | SMM | **0.72**s | **0.0551**% |
| | FFT | 0.78s | **0.0575**% |
| *Contracting-Expanding Distribution:* | | | |
| | SMM | **0.11**s | **0.184**% |
| FNO | Cubic Interpolation | 1.00s | 0.195% |
| | KB-NUFFT | 30s | 0.346% |
| | Toeplitz-NUFFT | 0.85s | 0.740% |
| **2D Lattice: Shear Layer** | | | |
| FNO | SMM | **57**s | **5.53**% |
| | Full Grid | 251s | 6.76% |
| UFNO | SMM | 68s | 5.61% |
| | Full Grid | 287s | 6.61% |
| FFNO | SMM | 108s | 12.4% |
| | Full Grid | 380s | 12.4% |
| **2D Lattice: Specific Humidity** | | | |
| FNO | SMM | **1.5**s | 4.65% |
| | Full Grid | 15s | 5.09% |
| UFNO | SMM | 2.5s | **3.91**% |
| | Full Grid | 23s | 4.34% |
| FFNO | SMM | 2.3s | 4.41% |
| | Full Grid | 26s | 5.02% |
| **2D Point Cloud: Flow Past Airfoil** | | | |
| FNO | SMM | 2.8s | **0.220**% |
| | Geometric Layer | 6.2s | 1.20% |
| UFNO | SMM | 2.6s | 0.380% |
| | Geometric Layer | 7.1s | 0.679% |
| FFNO | SMM | **2.1**s | 0.650% |
| | Geometric Layer | 6.9s | 2.10% |
| **2D Point Cloud: Elasticity** | | | |
| FNO | SMM | 0.41s | 1.96% |
| | Geometric Layer | 0.71s | 2.39% |
| UFNO | SMM | 0.60s | 2.05% |
| | Geometric Layer | 1.0s | 2.16% |
| FFNO | SMM | **0.28**s | **1.73**% |
| | Geometric Layer | 0.44s | 2.20% |
| **Random Spherical Point Cloud: Shallow Water Equations** | | | |
| | SMM | **15**s | **3.88**% |
| SFNO | Gaussian Interpolation | 86s | 7.29% |
| | Linear Interpolation | 71s | 12.7% |
| | SMM | 16s | 5.39% |
| FNO | Gaussian Interpolation | 92s | 8.41% |
| | Linear Interpolation | 83s | 15.2% |

To show the effectiveness as well as the generality of SMM we implement it within several prominent neural operators, namely the FNO, UFNO (Wen et al., 2022), the FFNO (Tran et al., 2023), and the SFNO (Bonev et al., 2023) (for data on the sphere). Model sizes are chosen to be as close as possible when comparing the SMM and the baselines. All experiments are performed on the Nvidia GeForce RTX 3090 with 24GB memory.

**Benchmark 1: Burgers' Equation.** The one-dimensional viscous Burgers' equation is a widely considered model problem for fluid flow given by

$$\partial_t u(x,t) + \partial_x(\frac{1}{2}u^2(x,t)) = \nu\partial_{xx}u(x,t) \qquad x \in (0,1) \quad t \in (0,1]$$
$$u(x,0) = u_0(x) \qquad x \in (0,1)$$

(8)

where $u$ denotes the fluid velocity and $\nu$ the viscosity. We follow (Li et al., 2020a) in fixing $\nu = 0.1$ and considering the operator that maps the initial data $u_0$ to the solution $u(\cdot, T)$ at final time $T = 1$. The training and test data, presented in (Li et al., 2020a) for this problem, is used. We start by comparing the standard version of FNO (with FFT) to FNO with SMM for data sampled on an equispaced grid. The differences between the resulting test errors is negligible, because the underlying algorithm is the same in this case modulo small numerical errors. Moreover, the training times per epoch were also comparable. In contrast, for data sampled from points drawn from a contracting-expanding distribution, illustrated in **SM** Figure 4, the proposed method was $5\%$ more accurate on

average when compared to FNO with a cubic interpolation, interpolating data from the contracting-expanding distribution to an equispaced grid. However, the training time with SMM was notably improved, by almost a factor of $4$, when compared to the interpolation baseline. Note that the cubic interpolation was computed beforehand and its cost not taken into account in reporting the training time. In addition, we draw comparisons with relevant Nonuniform FFT (NUFFT) algorithms with PyTorch implementations, namely the Kaiser-Bessel and Toeplitz NUFFT (Muckley et al., 2020).

**Benchmark 2: Shear Layer.** We follow a recent work on convolutional neural operators (Raonić et al., 2023) in considering the incompressible Navier-Stokes equations

$$\frac{\partial \mathbf{u}}{\partial t} + \mathbf{u} \cdot \nabla \mathbf{u} + \nabla p = \nu \Delta \mathbf{u}, \quad \nabla \cdot \mathbf{u} = 0. \tag{9}$$

Here, $\mathbf{u} \in \mathcal{R}^2$ is the fluid velocity and $p$ is the pressure. The underlying domain is the unit square with periodic boundary conditions and the viscosity $\nu = 4 \times 10^{-4}$, only applied to high-enough Fourier modes (those with amplitude $\geq 12$) to model fluid flow at *very high Reynolds-number*. The solution operator maps the initial conditions $\mathbf{u}(t = 0)$ to the solution at final time $T = 1$. We consider initial conditions representing the well-known *thin shear layer* problem (Bell et al., 1989; Lanthaler et al., 2021) (See (Raonić et al., 2023) for details), where the shear layer evolves via vortex shedding to a complex distribution of vortices (see **SM** Figure 10a for an example of the flow). The training and test samples are generated, with a spectral viscosity method (Lanthaler et al., 2021) of a fine resolution of $1024^2$ points, from an initial sinusoidal perturbation of the shear layer (Lanthaler et al., 2021), with layer thickness of 0.1 and 10 perturbation modes,the amplitude of each sampled uniformly from $[-1, 1]$ as suggested in (Raonić et al., 2023). As seen from **SM** Figure 10a, the flow shows interesting behavior with sharp gradients in two mixing regions, which are in the vicinity of the initial interfaces. However, the flow is nearly constant further away from this mixing region. Hence, we will consider input functions being sampled on a lattice shown in **SM** Figure 6 which is adapted to resolve regions with large gradients of the flow. On the other hand, the FFT based FNO, UFNO and FFNO baselines are tested on the equispaced point distribution. From Table 1, we observe that the proposed method is marginally more accurate while consistently being $4$ times faster per training epoch across all models, demonstrating a significant computational advantage on this benchmark.

**Benchmark 3: Surface-Level Specific Humidity.** Next, we focus on a *real world* data set and learning task where the objective is to predict the surface-level specific humidity over South America at a later time (6 hours into the future), given inputs such as wind speeds, precipitation, evaporation, and heat exchange at a given time. The exact list of inputs is given in **SM** Table 2. The physics of this problem are intriguingly complex, necessitating a *data-driven approach* to learn the underlying operator. To this end, we use the (MERRA-2) satellite data to forecast the surface-level specific humidity (EarthData, 2021 - 2023). Moreover, we are interested in a more accurate regional prediction, namely over the Amazon rainforest. Hence, for models with SMM, we will sample data on points on a lattice that is more dense over this rainforest, while being sparse (with smooth transitions) over the rest of the globe, see **SM** Figure 7 for visualization of this lattice. Test error is calculated over the region, shown in **SM** Figure 10b. The results, presented in Table 1, show that the localization capabilities of the proposed method not only offer more accurate results, but SMM based neural operators are also one order of magnitude faster to train than the baselines. The greater accuracy is also clearly observed in **SM** Figure 10b, where we observe that the FNO-SMM is able to capture elements such as the formation of vortices visible in the lower right hand corner and the mixing of airstreams over the Pacific Ocean.

**Benchmark 4: Flow Past Airfoil.** We also investigate transonic flow over an airfoil, as governed by the compressible Euler equations, with the problem setup considered in (Li et al., 2022), The underlying operator maps the airfoil shape to the pressure field. In this case, the underlying distribution of sample points changes between each input (airfoil shape). SMM-based neural operators can readily handle this situation. As baselines, we augment FNO, UFNO and FFNO with the geometric layer of Geo-FNO as proposed in (Li et al., 2022). To have a fair comparison with SMM based models, we allow Geometric layer based models to learn the underlying diffeomorphism online. The test errors, presented in Table 1 show that SMM-based models are both more accurate and much faster to train than the geometric layer based baselines.

**Benchmark 5: Elasticity-P.** Again, we follow (Li et al., 2022), to investigate the performance of this method on hyper-elastic materials. We take the data, exactly as outlined in their work. We also follow their training set up, using 1000 training data and 200 test data given as point clouds, predicting the stress as output. Again, we train all models (FNO, UFNO, FFNO) on the point cloud data with SMM and with the geometric layer (as proposed in (Li et al., 2022)) as a baseline. For each underlying model, the SMM is more accurate than the geometric layer while being significantly faster to train.

**Benchmark 6: Spherical Shallow Water Equations.** We follow the recent work on spherical neural operators (Bonev et al., 2023) by considering the shallow water equations on a rotating sphere. In this experiment, training data is generated on the fly such that new data is used for training and evaluation in every epoch, as described in (Bonev et al., 2023). From each data set, we draw points from a random uniform distribution over the sphere to create a point cloud, as opposed to a grid, see **SM** Figure 9 for an illustration of this point cloud. These may correspond to sensors over the globe in real-world applications. The training data for both models each have approximately 5000 points. As baselines, we considered SFNO but with data interpolated back to a regular grid on the sphere using either linear interpolation or radial basis function interpolation with a Gaussian kernel. We also compare the FNO on the interpolated data with the SMM approach on the point cloud data. The results, presented in Table 1, clearly show that SFNO using SMM readily outperforms baselines on the interpolated data in terms of both accuracy as well as training speed. This difference is very pronounced for the baselines based on interpolation, which are both expensive as well as inaccurate. As expected the FNO-SMM model performs significantly worse than the SFNO-SMM model which takes into account the underlying spherical geometry. Additionally, we provide the baseline comparison of the SFNO and FNO on the original (not from interpolation) grid in **SM** Table 5 as a reference to their relative performance.

Summarizing, the results of the numerical experiments clearly demonstrate that SMM is able to handle data sampled on arbitrary point distributions and accurately approximate the underlying operators. It readily outperforms baselines, based either on interpolation or geometric transformations, both on accuracy as well as on training speed. This is particularly evident on problems where the data is sampled on point clouds and where other operator learning methods such as U-Nets or convolutional neural operators cannot be readily applied.

## 4 DISCUSSION

**Summary.** Widely used neural operators such as FNO and its variants are limited to input/output data sampled on equispaced grids as the underlying spectral transformations can only be efficiently evaluated on this data structure. To expand the range of such neural operators, we propose replacing the standard computations of these spectral transformations within neural operators with a novel, simple and general method that leverages the Vandermonde structure of the underlying matrices to process data efficiently even on arbitrary sampling point distributions (point clouds). By design, our method is expected to process data on arbitrary point distributions with low computational cost. A novel PyTorch implementation is also presented to realize the proposed algorithm that we term as structured matrix method (SMM).

SMM is flexible enough to be embedded within any neural operator that uses non-local spectral transformations. We test with a variety of neural operators such as FNO, UFNO, FFNO and recently SFNO for data on a sphere. Moreover, SMM is general enough to handle point distributions ranging from equispaced grids through lattices to randomly distributed point clouds. We investigate the efficiency of the proposed SMM by testing it in conjunction with a variety of available neural operators on a suite of benchmarks that correspond to a variety of PDEs as well as sampling point distributions. In all the considered benchmarks, SMM outperformed the baselines (based either on interpolation or learnable geometric diffeomorphisms as proposed in (Li et al., 2022)) both in terms of accuracy as well as training speed, even showing order of magnitude speedups in some cases. Additionally, we present in **SM** Table 6, the generalization capabilities of the SMM approach to be applied to point clouds not encountered during training time, even when the number of points varies greatly. Thus, we present a novel yet simple method that can be used readily within neural operators to handle input/output data sampled on arbitrary point distributions and learn the underlying operators accurately and efficiently.

**Related Work.** We start with a succinct summary of the extensive literature on Vandermonde-structured matrices and related constructions. In this context, the delay Vandermonde matrix (DVM) is a superclass of the DFT matrix. The DVM structure is utilized to analyze TTD wideband multi-beam beamforming while solving the longstanding beam squint problem (Perera et al., 2022; 2018; Ariyarathna et al., 2017; Perera et al., 2020a;b). Although the Vandermonde matrices can be ill-conditioned (Walter, 1975; Higham, 2002; 1996; 1987; Demmel & Koev, 2005; Pan, 2001; 2016), the proposed SMM does not encounter ill-conditioning, as nodes are placed on the unit disk (Pan, 2016; Perera et al., 2021; 2020b), and we do not compute the explicit inverse of the Vandermonde-structured matrix because it is too expensive and numerically less accurate (Higham, 1996). Nonequispaced FFT (NUFFT) has already been developed (Dutt & Rokhlin, 1993; Beylkin, 1995; Dutt & Rokhlin, 1995; Liu & Nguyen, 1998; Kircheis & Potts, 2023) as well as libraries for efficient GPU implementations (Shih et al., 2021) and PyTorch implementations (Muckley et al., 2020). The techniques frequently used in these NUFFTs or approximated inverse NUFFT include a combination of interpolation, windowing, and sampling along with FFT to achieve $\mathcal{O}(n \log n)$ complexity (Selva, 2018; Heinig & Rost, 1984; Gelb & Song, 2014; Kunis & Potts, 2007; Ruiz-Antolin & Townsend, 2018; Averbuch et al., 2016; Kircheis & Potts, 2019; 2018; Kunis, 2006). Essentially, the NUFFT involves interpolating to the equispaced grid followed by the FFT. An interpolation approach has already been shown to be sub-optimal for the case of neural operator learning (Li et al., 2022). Furthermore, interpolation within the operator structure increases the computation time, making it more efficient to interpolate data to a grid before training or testing and simply apply the standard FNO. The nonequispaced DFT is also presented as a summation (Fessler & Sutton, 2003; Liu & Nguyen, 1998; Kircheis & Potts, 2023), which can be arranged into a Vandermonde-structured matrix, but this is not presented in the literature. Excluding fast transforms, the structure of the Vandermonde matrix has been employed to perform the nonequispaced, or nonuniform discrete Fourier transform (NUDFT) (Bagchi & Mitra, 1999) in 1D and 2D; however, the two-dimensional distributions, in this case, are limited to the lattice and nonuniform parallel lines. The methods we propose in this paper extend the use of Vandermonde-structured matrices to irregular point distributions in two or more dimensions. The NUDFT is rarely used, as many applications of Fourier transforms require all Fourier coefficients, resulting in $O(n^2)$ cost (Bagchi & Mitra, 1999). However, this is not the case for the neural operators that we consider, and thus the proposed SMM avoids the rapidly growing computational costs associated with NUDFT. In some implementations, the Geo-FNO also uses a nonequispaced Fourier transform in conjunction with the diffeomorphism. As opposed to constructing a standard matrix, (Li et al., 2022) construct an $N + 1$ tensor, $N$ being the number of spatial dimensions. This geometric-Fourier layer then transforms the problem to a set of points on a regular grid, similar to a sinc interpolation. From this point, the FFT is used. While the nonequispaced transformation does have some similarities to that proposed here, the proposed method differs fundamentally in its use, leading to notable differences in the results. The construction and multiplication we offer in our implementation are fast and efficient, allowing us to use the nonequispaced transformation within each Fourier layer, as opposed to transforming the data to an equispaced grid through some slower transformation process and then apply the FFT. Likewise, this method allows us to avoid the need for any sort of diffeomorphism. In practice, we found the diffeomorphism layer difficult to train in tandem with the Fourier layer of the FNO. This process often requires careful tuning of the associated diffeomorphism loss, as well as the need to freeze this layer after some point of time and to retrain the FNO with the diffeomorphism model fixed. In constrast, the FNO-SMM converges just as reliably as the basic FNO and we see the same behavior with variants of FNO such as UFNO and FFNO.

**Limitations and Future Work.** The elements of the Vandermonde-structured matrix are directly related to the positions of the data points for a given problem. Thus, if all training samples have different point clouds, a different matrix must be constructed for each sample. Constructing the matrices at run time, *i.e.*, during training, hinders performance; however, this is mitigated by the tensorized matrix construction methods available in the implementation. As outlined by the results, the run-time matrix construction is still able to outperform other techniques for handling point cloud data. Finally, we would like to highlight the potential of the SMM method for dealing with very general spectral transformations for data sampled on arbitrary point distributions. We have considered both Fourier transforms and Spherical harmonics in this paper but other relevant spectral transforms such as Wavelets or Laplace transforms will also be considered in future work.

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

**Supplementary Material for:**
A Structured Matrix Method for Nonequispaced Neural Operators.

# A   Technical Details

## A.1   The Fourier Neural Operator.

Operator learning is concerned with learning the mapping between separable Banach spaces, typicall denoted as $\mathcal{A} = \mathcal{A}(D; \mathbb{R}^{d_a})$ and $\mathcal{U} = \mathcal{U}(D; \mathbb{R}^{d_u})$, where $D \subset \mathbb{R}^d$ is a bounded, open set. The mapping is denoted as $G^{\dagger} : \mathcal{A} \to \mathcal{U}$. With observations $\{a_j, u_j\}_{j=}^{N}, a_j \sim \mu, u_j = G^{\dagger}(a_j)$, the operator $G^{\dagger}$ may be approximated by map parameterized by a neural network, $G_{\theta}, \theta \in \Theta$, where $\Theta$ is a finite-dimensional parameter space. The operator is then approximated by minimizing a cost functional

$$\min_{\theta \in \Theta} \mathbb{E}_{a \sim \mu} \left[ C(G(a, \theta), G^{\dagger}(a)) \right] \tag{10}$$

Briefly, the critical component of operator learning is the *kernel integral operator* $\mathcal{K}$,

$$\left( \mathcal{K}(a, \phi) v_t \right)(x) := \int_D \kappa(x, y, a(x), a(y); \phi) v_t(y) \mathrm{d}y, \forall x \in D, \tag{11}$$

where $\kappa_{\phi} : \mathbb{R}^{2(d+d_a)} \to \mathbb{R}^{d_v \times d_v}$ is a neural network parameterized by $\phi \in \Theta_{\mathcal{K}}$, $v_t$ is an iteration of a sequence of functions each taking values in $\mathbb{R}^{d_v}$, and $x, y$ correspond to measurement locations in the given sample. This kernel function is to be learned from the data; however, computing the integral is prohibitively expensive for large data sets. This problem is mitigated by the convolution property of the Fourier transform.

The kernel integral operator may be redefined within the Fourier space as

$$\left( \mathcal{K}(a, \phi) v_t \right)(x) = \mathcal{F}^{-1} \left( R_{\phi} \cdot (\mathcal{F} v_t) \right)(x), \forall x \in D, \tag{12}$$

where $\mathcal{F}$ denotes the Fourier transform of a function, $\mathcal{F}^{-1}$ its inverse, and $R_{\phi}$ is the Fourier transform of a periodic function $\kappa : \bar{D} \to \mathbb{R}^{d_v \times d_v}$, parameterized by $\phi \in \Theta_{\mathcal{K}}$. By learning the kernel in the Fourier space, we avoid computing the integral. The computational complexity is then reduced from $O(n^2)$ to $O(n \log n)$ when using the FFT. For more details on the Fourier neural operator and neural operator learning, we refer the reader to the works by Li et al. Li et al. (2020a;b).

## A.2   The FFT Signal Flow Graph.

The signal flow graph provides a graphical representation of the FFT algorithm.

## A.3   On Computational Complexity of the Structured Matrix Method.

The Vandermonde matrix-vector product, as described in Section 2 of the main text, is broken down into a vector with $n$ components via inner products of vectors, requiring $2n$ real multiplications and $2(n-1)$ real additions having a real-valued input, a total of $4n^2 - 2n$ flops. In the implementation, the number of rows is reduced from $n$ to a constant $m$ s.t. $m << n$, as determined by the number of modes chosen for a given problem. Thus, for the 1D case, the total number of flops is $4nm - 4n$, therefore only growing linearly with the problem size. For the $d$-dimensional nonequispaced data, we construct a matrix with size $n$ columns but the number of rows $m_{total} = (m_1)(m_2) \ldots (m_d)$, where $m_j$ is the number of modes taken along the $j^{th}$ spatial dimension. Thus, the complexity of the proposed transformation using the Vandermonde-structure is reduced from $\mathcal{O}(n^2)$ to $\mathcal{O}(m_{total} n)$. Finally, we note here that the existing non-uniform FFT has the complexity of order $\mathcal{O}(Rn \log(n))$, where $R$ is based on diagonally scaled FFT. Thus, applying the existing non-uniform FFT on FNO won't reduce the complexity as of the proposed Vandermonde-structured approach on the neural operator with the complexity reduction of order $\mathcal{O}(m_{total} n)$.

For the extension to the spherical harmonics, we pre-compute the Legendre polynomials over $n$ points as an evaluation problem with $\mathcal{O}(n)$ complexity. Once these are pre-computed, we call these polynomials to calculate the spherical harmonics with complexity $\mathcal{O}(l^2)$, e.g. the order $l$ is calculated using an odd number of harmonics from 1 to the order $l$. Thus, the associated spherical

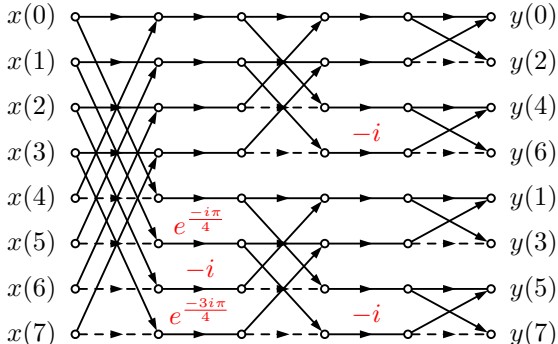

Figure 2: The 8-point fast Fourier transform signal flow graph. $x$ and $y$ represent the signal in the physical and Fourier domain, respectively. Dashed lines represent a multiplication by -1, red elements denote a multiplication by that factor, and converging arrows represent a sum.

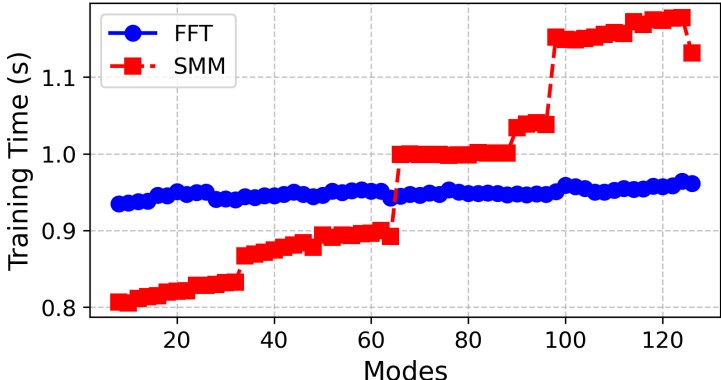

Figure 3: Ablation study to compare the FNO training time using the FFT and the proposed method for the 1D Burgers' equation on equispaced data. Even in the equispaced case, where FFT is applicable, the SMM method offers small, but clear, time well beyond the number of modes ($12 \leq m \leq 20$) typically used in neural operator learning. The FFT offers an advantage in the uniform case using 64 modes or greater.

harmonics can be computed using $\mathcal{O}(n\,l^2)$ complexity, where $l << n$. Finally, arranging harmonics into Vandermonde-structured form followed by the matrix-vector products into vector forms (as described above) reduces the complexity to $\mathcal{O}(nl^2)$ as opposed to $\mathcal{O}(n^2\,l^3)$.

The question may arise; *Can the number of modes really be said to be constant with respect to the number of points, as more points may be used for more complex problems, necessitating more modes?* A priori, it is not possible to say how many modes must be selected for a problem of given complexity or resolution - these must be determined through model selection, varying the number of modes, and choosing the model that minimizes the error over the validation set. This model selection process has revealed that there is often a point where increasing the number of modes results in worse performance, i.e. increasing the number of modes will not automatically result in increased performance. With regard to the sharp features and large gradients that may arise in complex sets of PDEs or high-resolution data, the lifting layer and the skip connections of the model architecture are much more capable of resolving such features than the Fourier layers. The Fourier layers serve to propagate information throughout the domain. Thus, we maintain the position that the number of modes is fixed with respect to the number of points.

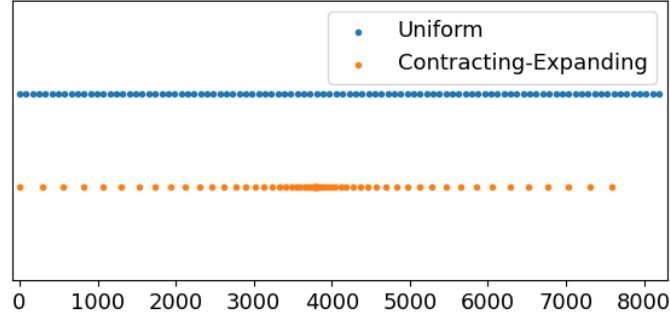

Figure 4: Point distributions used in the Burgers' equation experiments. Data is selected from the uniform distribution to construct the contracting-expanding distribution and random distribution. The space between points in the contracting-expanding distribution grows from a point in both directions according to a geometric distribution.

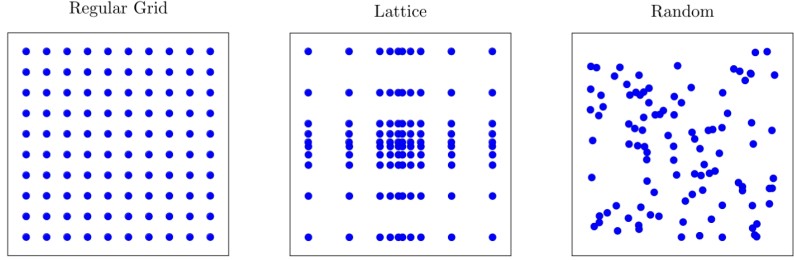

Figure 5: Distributions discussed in this paper. The FNO is restricted to the regular grid. The structured matrix method may be applied to the lattice distribution via equation 3, or the random distribution via equation 4

### A.4 POINT DISTRIBUTIONS

We investigate both a uniform and a *contracting-expanding* distribution to help lay the foundation for the proposed method. These distributions are visualized in Figure 4, which also shows a random distribution.

For the two dimensional experiments, we investigate lattices as well as random or structured data on a point cloud, simplified illustrations of which are provided in Figure 5. A lattice with a nonuniform distribution along one axis (*Shear Layer*) and a lattice with a nonuniform distribution along both axes (*Surface-Level Specific Humidity*) are investigated and visualized in Figures 6 and 7. Point cloud data with varying geometries are investigated in the *Airfoil* and *Elasticity* experiments. We provide visualizations of several airfoil point clouds in Figure 8.

Finally, we investigate the performance of spherical harmonics models on a random distribution over the sphere. The original point cloud, the random points, and the grid generated by interpolating from the random points are shown in Figure 9.

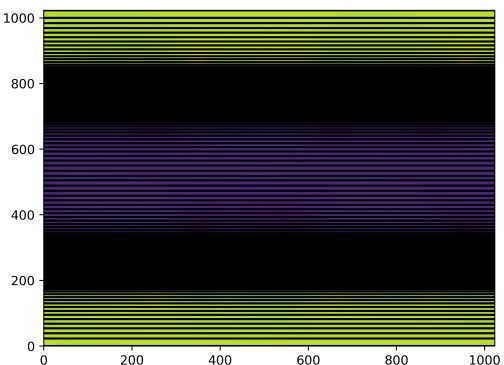

Figure 6: Nonequispaced lattice for the shear layer problem. Sampling is dense close the interface region, smoothly becoming sparse further from this region.

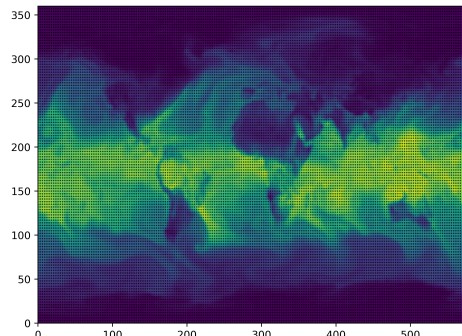

(a) FNO point distribution. The points displayed in this image have been subsampled from the original distribution to maintain clarity in the figure.

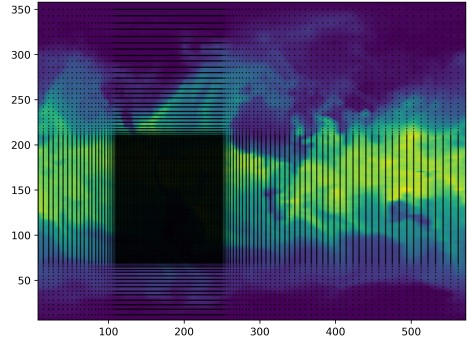

(b) Structure matrix method point distribution. A densely sampled region is located over South America and the lattice becomes more sparse further from this region.

Figure 7: Distributions used within the surface-level specific humidity experiment.

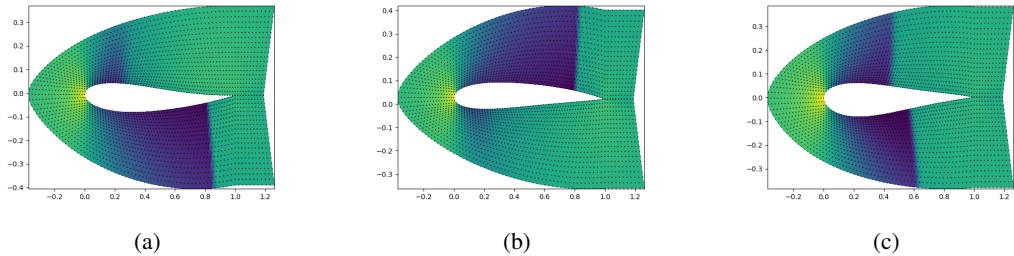

Figure 8: Several point distributions around airfoils and their associated pressure distributions.

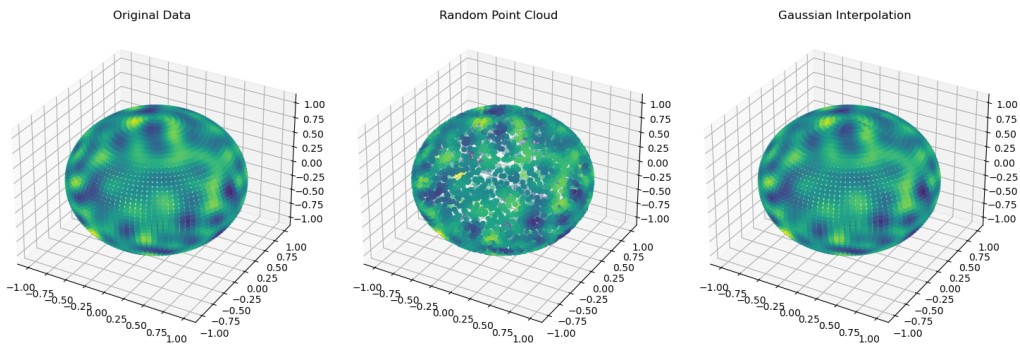

Figure 9: The original distribution of points (left) is sampled from at locations chosen randomly along the surface of a sphere (middle). To compare with baselines, which require a uniform distribution, we use a radial-basis function interpolation scheme with a Gaussian kernel and a variance of 0.1 to interpolate from the random points back up to a grid (right).

## A.5   Additional Training Details and Results.

We use 16 different parameters to make predictions for the *Surface-Level Specific Humidity* experiments. These are listed in this subsection in Table 2. We also show the predictions of several models for several experiments in Figure 10. Table 3 provides information on the distributions of the minimum median relative L1 test error given different initial seeds. We also provide histograms showing the error for all test samples in Figures 11 and 12. Finally, we include Table 4 to show the sizes of the various models.

## A.6   Generalization Capabilities for the SWE Experiments.

First, we provide the errors of the FNO and SFNO on their respective grids for the Shallow Water Equations experiments on the Sphere in Table 5. We find that the training times are approximately 30 seconds, twice that of the SMM approaches on the unstructured data, and the errors over the collocation points are equivalent to those of the SMM approach.

We also present the results for the SFNO-SMM on new sets of points which are unseen at training time. We vary the number of points, providing a sensitivity analysis. The results, in Table 6, show that the error remains constant.

Table 2: Inputs for the surface level specific humidity predictions.

| Acronym in MERRA-2 | Input | Units |
|---|---|---|
| CDH | heat exchange coefficient | $\frac{kg}{m^2 s}$ |
| CDQ | moisture exchange coefficient | $\frac{kg}{m^2 s}$ |
| EFLUX | total latent energy flux | $\frac{W}{m^2}$ |
| EVAP | evaporation from turbulence | $\frac{kg}{m^2 s}$ |
| FRCAN | areal fraction of anvil showers | $1$ |
| FRCCN | areal fraction of convective showe | $1$ |
| FRCLS | areal fraction of large scale show | $1$ |
| HLML | surface level height | $m$ |
| QLML | surface level specific humidity | $1$ |
| QSTAR | surface moisture scale | $\frac{kg}{kg}$ |
| SPEED | surface wind speed | $\frac{m}{s^2}$ |
| TAUX | eastward surface stress | $\frac{N}{m^2}$ |
| TAUY | northward surface stress | $\frac{N}{m^2}$ |
| TLML | surface air temperature | $K$ |
| ULML | surface eastward wind | $\frac{m}{s}$ |
| VLML | surface northward wind | $\frac{m}{s}$ |

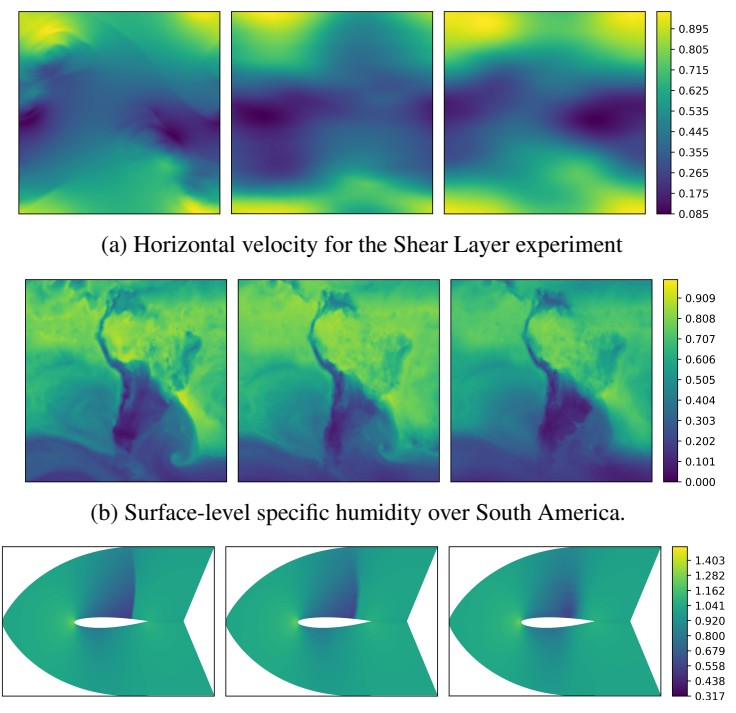

(a) Horizontal velocity for the Shear Layer experiment

(b) Surface-level specific humidity over South America.

(c) Pressure distributions around an airfoil.

Figure 10: These figures display examples of the ground truth, the target which the FNO-SMM, FNO, or Geo-FNO attempt to match. Left: Ground Truth. Center: FNO-SMM Right: FNO for (a) and (b) and Geo-FNO for (c).

Table 3: The mean and standard deviation of the median test errors over 10 different initializations of the structured matrix method for the *Burgers'* experiments.

| Data Distribution | Model | Method | Mean±Std |
|---|---|---|---|
| *Uniform* | FNO | FFT | $0.0650 \pm 0.0051\%$ |
| *Uniform* | FNO | SMM | $0.0653 \pm 0.0074\%$ |
| *Contracting-Expanding* | FNO | FFT+Interpolation | $0.2076 \pm 0.0087\%$ |
| *Contracting-Expanding* | FNO | SMM | $0.1973 \pm 0.0069\%$ |

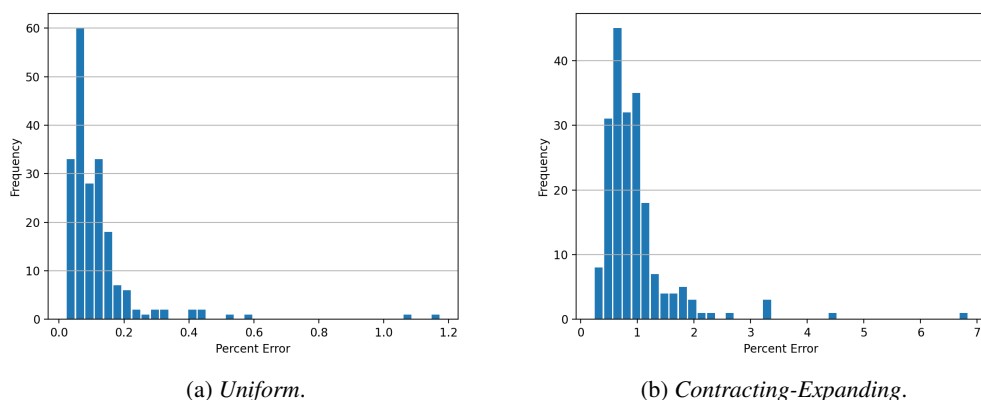

(a) *Uniform.*          (b) *Contracting-Expanding.*

Figure 11: Histograms displaying the distribution of test errors for the different point distributions of the *Burgers'* numerical experiments using the structured matrix method (SMM).

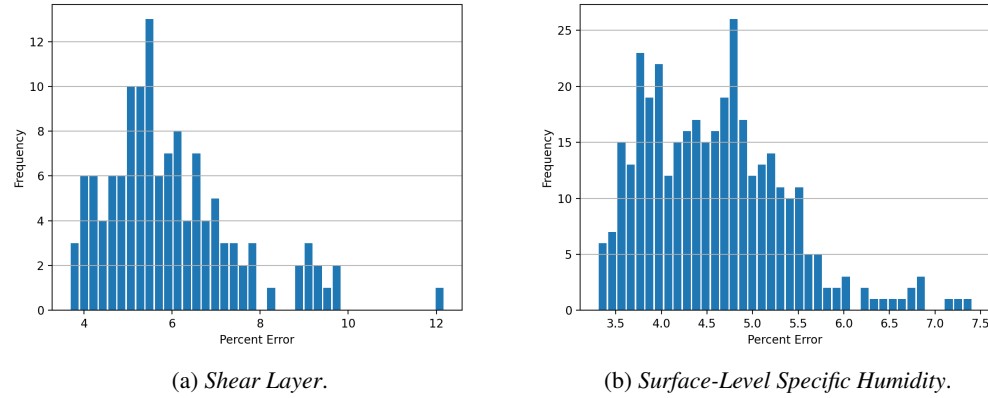

(a) *Shear Layer.*        (b) *Surface-Level Specific Humidity.*

Figure 12: Histograms displaying the distribution of test errors for the two-dimensional numerical experiments on the lattice using the FNO-SMM model and method combination.

Table 4: Model sizes in terms of the number of parameters. Within an experiment, the models are constructed to have similarly sized convolution and spectral convolution layers. This results in the FFNO often having an order of magnitude fewer parameters, a key contribution outlined by Tran et al. (2023). The UFNO models have the most parameters due to the extra U-Net layers, and the Geometric Layer results in a slightly higher number of parameters as well.

| Model | Method | No. Parameters | No. Modes |
|---|---|---|---|
| **1D: Burgers' Equation** | | | |
| *Equispaced Distribution:* | | | |
| FNO | SMM | 287425 | 16 |
| | FFT | 287425 | 16 |
| *Contracting-Expanding Distribution:* | | | |
| FNO | SMM | 287425 | 16 |
| | Interpolation | 287425 | 16 |
| **2D Lattice: Shear Layer** | | | |
| FNO | SMM | 3162881 | 20 |
| | Full Grid | 3162881 | 20 |
| UFNO | SMM | 3750401 | 20 |
| | Full Grid | 3750401 | 20 |
| FFNO | SMM | 423041 | 20 |
| | Full Grid | 423041 | 20 |
| **2D Lattice: Specific Humidity** | | | |
| FNO | SMM | 8398049 | 32 |
| | Full Grid | 8398049 | 32 |
| UFNO | SMM | 8691553 | 32 |
| | Full Grid | 8691553 | 32 |
| FFNO | SMM | 537697 | 32 |
| | Full Grid | 537697 | 32 |
| **2D Point Cloud: Flow Past Airfoil** | | | |
| FNO | SMM | 1188577 | 12 |
| | Geometric Layer | 1250339 | 12 |
| UFNO | SMM | 1482081 | 12 |
| | Geometric Layer | 1840163 | 12 |
| FFNO | SMM | 500673 | 14 |
| | Geometric Layer | 526787 | 14 |
| **2D Point Cloud: Elasticity** | | | |
| FNO | SMM | 1484289 | 12 |
| | Geometric Layer | 1546403 | 12 |
| UFNO | SMM | 1778049 | 12 |
| | Geometric Layer | 1840163 | 12 |
| FFNO | SMM | 526787 | 14 |
| | Geometric Layer | 533441 | 14 |
| **Random Spherical Point Cloud: Shallow Water Equations** | | | |
| SFNO | SMM | 39749251 | $l = 22$ |
| | Gaussian Interpolation | 39201536 | $l = 22$ |
| | Linear Interpolation | 39201536 | $l = 22$ |
| FNO | SMM | 31978563 | 20 |
| | Gaussian Interpolation | 39201536 | 20 |
| | Linear Interpolation | 39201536 | 20 |

| Method | Training Time (per epoch) | Error over Collocation Points |
|--------|---------------------------|-------------------------------|
| SFNO | 30s | 3.60% |
| FNO | 32s | 5.74% |

Table 5: Accuracy of the SFNO and FNO when trained on the original $256 \times 512$ grid which has been sub-sampled to a $51 \times 102$ grid. At inference time, we calculate the error over the collocation points, i.e. those used in the random distribution.

| Approximate No. Points | Error |
|------------------------|-------|
| 4000 | 6.36% |
| 8000 | 5.64% |
| 15000 | 5.44% |
| 29000 | 5.39% |
| 50000 | 5.43% |

Table 6: Accuracy of the SFNO-SMM on test sets of the Spherical Shallow Water Equations experiment with new random distributions of varying size. Each test set consists of 64 samples. The model was trained on data sets of approximately 5000 points. All point distributions and test samples used in this evaluation were not seen during training. The results show that the proposed method is able to generalize to new point distributions.

