# OpenReview forum: "A Structured Matrix Method for Nonequispaced Neural Operators"
_ICLR.cc/2024/Conference — Submitted to ICLR 2024_

### Official Review · Reviewer_7coN · 2023-10-30

**Soundness:** 1 poor
**Presentation:** 2 fair
**Contribution:** 1 poor
**Rating:** 5
**Confidence:** 4

**Summary:**

This manuscript proposes that when building Fourier Neural Operators (FNOs) that have unequally spaced sample points and relatively few "modes," it is faster to simply evaluate the exponential sums directly rather than use a fast transform. The efficacy of such a method is supported by several numerical experiments.

**Strengths:**

The manuscript considers experiments with a relatively broad range of models and does show that if few modes are considered then evaluating the exponential sums directly is likely faster. An implementation is provided.

**Weaknesses:**

The primary weakness of this manuscript is in its framed contribution and presentation around the "structured matrix method" (with a secondary issue being the baselines used and some details of the numerical experiments).

For the so-called Type 1 NUFFTs the authors consider, the fact that direct evaluation of the exponential sums is more efficient if only a small number of modes are needed is well known; in fact the same comment applies to the equispaced transform as well (albeit with a different crossover point with respect to the number of modes). In fact, standard NUFFT libraries make this clear, see, e.g., the "Do I even need a NUFFT?" section of https://finufft.readthedocs.io/en/latest/. Therefore, it is not clear what the contribution is here. Moreover, the presentation of the method in Section 2 is not even particularly clear (especially when moving to general point clouds since the notation seems to reduce to the case with only 2 sampling points? There are much nicer ways to write the transform).

The section titled "Inverse Transformations" is misleading. The manuscript seems to imply the adjoint transform is what is meant—these are not the same thing. (This is used incorrectly throughout the text.) On this point, the discussion in the related work section talking about the lack of a "direct" inverse NUFFT is also misleading. In particular Type 1 and Type 2 NUFFTs are not inverses of each other (unlike in the equispaced case) and the "inverse" refereed to is in, e.g., the least squares sense for things like imaging problems; this is not something the authors provide.

Compounding this issue, the discussion of the computational complexity is lacking. The complexity should always be written in terms of the number of modes and the number of sample points (e.g., $\mathcal{O}(mn)$ with $n$ sample points and $m$ modes)—the claim could then be made that $m$ is often "constant" as $n$ grows and in that situation the complexity is linear. (Though, this is likely a bit disingenuous since if the underlying problem got more complicated, rather than simply oversampling a simple problem, both $m$ and $n$ would likely have to grow.) Given the stress placed on the number of modes it is also somewhat surprising that, unless I missed it, the experiments do not talk about how many modes are used in each case (that seems easy to add and informative).

For the experiments it is also not clear why something like https://finufft.readthedocs.io/en/latest/ is not used as a comparison point (rather than cubic interpolation). Moreover, the "full grid" comparisons seem misleading as well; if I am understanding this point correctly the "Full Grid" baseline is using more modes rather than the reduced number. If so, shouldn't it be done with the reduced number and, if that number is small enough, directly evaluate the sums? Also,

Lastly, some rather relevant references are missing

Related to the software references above:

A parallel non-uniform fast Fourier transform library based on an “exponential of semicircle” kernel. A. H. Barnett, J. F. Magland, and L. af Klinteberg. SIAM J. Sci. Comput. 41(5), C479-C504 (2019).

These two are some of the first formalizations of NUFFTs, so they seem relevant (certainly more so than many of the other included references):

Dutt, Alok, and Vladimir Rokhlin. "Fast Fourier transforms for nonequispaced data." SIAM Journal on Scientific computing 14.6 (1993): 1368-1393.

Beylkin, Gregory. "On the fast Fourier transform of functions with singularities." Applied and Computational Harmonic Analysis 2.4 (1995): 363-381.

**update after author response**

I would like to thank the authors for their thoughtful responses to my concerns and those of the other reviewers.

In brief, I completely agree with the authors (and other reviewers) that the main contribution of this work is, in some sense, a comprehensive experimental illustration that for certain FNO problems a small enough number of non-equispaced modes are needed that direct evaluation of exponential sums is sufficiently efficient. (I should have been clear that anytime I say/said “evaluation of sum” I am implicitly implying as matrix/tensor products—as would be standard for anyone working on these fast transforms given that the matrix forms are well known/understood).

However, the initial framing of the manuscript (and the current version in some respects—look at the title and framing “question” for the contributions) strongly suggests that the contribution is in the evaluation of spectral transforms. And, as noted in my initial review, I don’t see any “algorithmic” contribution in that regard (though, I agree that an implementation is nice). For example, the statement in one of the replies about the “full grid”:

	“The number of modes is indeed low enough for the sum to be directly evaluated, specifically through a PyTorch matrix multiplication function, and this is one of our primary contributions in this paper.”

Is not something I see as a contribution—given how small $m$ is that is how it would/should have always been done.

From my perspective the paper as written is a bit of a red-herring with regards to its contribution. It seems that a more appropriate framing would actually be “a realization of FNOs on non-uniform grids” and when the transforms are discussed there would just be a simple/short section noting that the number of needed modes is small enough that the transforms can be evaluated efficiently using matrix/tensor constructions. As structured the manuscript still suggests a contribution in “structured matrix methods” that is useful for FNOs. Similarly, given how small $m$ is the NUFFT comparisons are somewhat superfluous/uninteresting (see the FAQ linked to in my original review) as is the long “related work” section (given the contribution isn’t the relevant related work FNOs that use full grids?). Nevertheless, I do think they are nice to see and fine to include.

Ultimately, I think that my difference of opinion with the other reviewers may be partially explained by (primary) background/area. From the FNO perspective there are interesting results here. However, from the “structured transforms” perspective it is completely unclear why there was a gap between the promise and delivery of using arbitrary grids given that direct evaluation was/is always an option that is likely the best choice given the context (this also holds for the other transforms mentioned, again this makes saying the method has “broader scope” a bit odd). Again, see the link in my initial review—the pursuit of NUFFTs or even the use of FFTs was in some ways a red-herring given the size of $m.$

All those thoughts aside, I have slightly updated my score. I still think that the manuscript is presented as the contribution somewhat being the transforms and I just don’t see one there beyond the implementation. I will leave it to the area chair to determine if they feel the current version of the manuscript is a suitable match for the agreed upon contribution.

**Questions:**

For the "Full Grid" baseline, is the model using more modes rather than the reduced number. If so, shouldn't it be done with the reduced number and, if that number is small enough, directly evaluate the sums?

---

> ### Author Response · Authors · 2023-11-17
> **Response to Reviewer 7coN**
>
> We start by thanking the reviewer for your appreciation of the merits of our paper and your welcome suggestions to improve it. Below, we address the concerns raised by the reviewer and thank the reviewer in advance for their patience in reading our detailed reply.
>
> 1. We start by addressing the reviewer's concerns about **framed contribution of the article and presentation around the "structured matrix method"**: In this regard, we would like to point out a possible misunderstanding about our contributions. While we agree completely with the reviewer that there has been considerable body of work on *non-uniform FFTs* and related algorithms in the signal processing literature, among others, our aim in the paper was NOT to propose a new method for evaluating the discrete Fourier transform efficiently on non-uniform grids. Rather, our goal was to propose a computationally efficient realization of widely popular neural operator architectures, such as FNO, which are currently restricted to uniform grids, on arbitrary point clouds. To this end, we needed to replace the FFT module inside the Fourier layers of FNO with an alternative that can be computationally efficient on unstructured grids. We do so by leveraging the following:
>     i) the observation, backed by both empirical work as in the original FNO paper as well as more recently by theory (see ArXiv:2210.01074 and ArXiv:2304.13221), that FNO type models only require a fixed small (compared to the number of points) number of Fourier modes for accurate operator approximation
>     ii) the DFT structure, being a special Vandermonde matrix, makes it relatively straightforward to obtain a fast algorithm by directly evaluating exponential sums
>     iii) availability of highly optimized functions within PyTorch such as *torch.bmm* and *torch.matmul* for tensor operations that allows us to computationally realize the fast DFT on arbitrary point clouds.
>     We combine all these ingredients into the SMM algorithm and demonstrate through considerable empirical evaluation that it provides a simple, accurate and fast realization of neural operators on arbitrary point clouds. This is the main contribution of this paper and it has been acknowledged as such by other reviewers. We have modified the main text to more explicitly frame our contributions and to mention that we do not propose a new non-uniform FFT algorithm but rather employ a suitable one within the framework of operator learning. This should also be contrasted with existing algorithms that extend neural operators to arbitrary grids. These approaches rely on either interpolation to regular grids or learnable geometric transformations between point clouds and regular grids, making them difficult to train and generalize and/or inaccurate. We also demonstrate through our experiments that our approach leads to a significant improvement over the state of the art methods in this regard (see Table 1).
>
> 2. With respect to the reviewers' comments on the **experiments, comparisons, and existing NUFFT libraries**: we thank the reviewer for pointing this out. Following your excellent suggestion, we have now compared our approach with existing NUFFT libraries. Of course, to apply these neural operators to nonequispaced data, any type-1 NUFFT could be a valid approach, as it is the FFT which limits the neural operator to the equispaced regime. However, we require not only a GPU implementation but also support within the NUFFT for *backpropagation* in order to train the neural operators. Therefore, the PyCuFiNufft of the referenced library cannot be used for this purpose. Given this, we opt for the TorchKbNufft library, which offers both the Kaiser-Bessel and Toeplitz NUFFT algorithms with support for deep learning architectures. We have repeated the Burgers' experiment on the nonequispaced data with these NUFFT algorithms and present the results in the Table below (see also Table 1 of the main text).
>
> | Method          | Training Time (per epoch) | Error  |
> |-----------------|---------------------------|--------|
> | SMM             | 0.11s                     | 0.184% |
> | KB NUFFT        | 30s                       | 0.35%  |
> | Toeplitz NUFFT  | 0.85s                     | 0.74%  |

---

> ### Author Response · Authors · 2023-11-17
> **Response to Reviewer 7coN continued.**
>
> The results show that our approach clearly and significantly outperforms these NUFFT algorithms in both speed and accuracy. The KB NUFFT uses an interpolation step before applying the FFT, and the Toeplitz NUFFT corrects imperfections in the frequency domain by using a scaling matrix. Because these algorithms will be called several thousands of times within the training process, it is more efficient to do the interpolation beforehand, as opposed to in the loop. Likewise, quicker algorithms like the Toeplitz NUFFT will introduce errors greater than those of interpolation. These observations might very well have been the reason why NUFFT approaches have not been considered in the context of operator learning before and the preference has been towards interpolation approaches. Nevertheless, we thank the reviewer again for asking us to compare with NUFFT as we believe that it further strengthens the rationale and contributions of our paper. We have also explicitly mentioned this point in the main text of the revised version now.
>
> 3. With regards to the reviewers' comments on **terminology and transform representations**, we start by acknowledging that some terms were indeed used loosely and imprecisely in our original manuscript. In particular, the adjoint and the inverse are indeed not equivalent in the nonuniform case. This was an abuse of terminology on our part. For uniform discretizations, the inverse DFT is used for the Fourier neural operator in the discrete case, and we chose terminology to reflect this practice. To clarify the underlying process, we compute the product of an integral kernel matrix and the data in the frequency space, where we then use the conjugate transpose of the Vandermonde-structured matrix to transform our data back from the frequency domain to the spatial domain. The Vandermonde-structured matrices have an extremely ill-conditioned inverse, which is why explicit computation of the inverse of Vandermonde-structured matrices is rarely done. Thus, to be more clear to the reader, we have now changed the term "inverse transformation" to the "backward transformation" to denote the mapping back from frequency to physical space.
>
> To maintain clarity, we also have corrected a minor typo in the section "Extension to N-dimensional Point Clouds." We intended to describe the particular example of the two-dimensional point cloud of $n$ points. In this case, the data should be represented as $X\in \mathbb{R}^{n}$ or $\mathbb{C}^{n}$, while the positions are stored as $P \left[ \boldsymbol{p}_0, \boldsymbol{p}_1 \right]\in \mathbb{R}^{n \times 2}$, the first and second columns being associated with the positions of point along the first and second spatial axes, respectively. This is not to say that the samples are reduced to simply 2 points, but rather that we have $n$ sample points whose positions are defined by their locations along the two spatial dimensions.
>
> Furthermore, we agree that there are nicer ways to write such transforms, such as through summation notation, which is often done in literature. However, as the implementation of this transform within the context of machine learning through a PyTorch library is of special interest, we choose to describe the matrices themselves, as opposed to the transform. For problems where points vary between training samples, it is also paramount that these matrices may be efficiently constructed, mainly by avoiding **for loops**. Therefore, we hope to provide a general illustration of the structure of the matrix itself, as well as the implementation available in the provided code so that others planning to implement FNO-based methods on nonequispaced point-clouds will be able to tailor such an approach to their specific application.

---

> > ### Author Response · Authors · 2023-11-17
> > **Response to Reviewer 7coN continued.**
> >
> > 4. In response to the reviewer's comments on **computational complexity and model specifications**, we agree it is more precise to include the number of modes in the computational complexity, and we have made this update in the revised version (See **SM** Section A.3). We have also included the number of modes for each benchmark in **SM** Table 4. As observed from the table, the number of Fourier modes is always in the range $12-32$ and is significantly less than the number of collocation points.
> >
> > The reviewer also raises an interesting question regarding the empirical relations between the number of modes and the number of points. A priori, it is not possible to say how many modes must be selected for a problem of given complexity or resolution - these must be determined through model selection, varying the number of modes, and choosing the model that minimizes the error over the validation set. This model selection process generally reveals that there is a point where increasing the number of modes results in worse performance for a given data set. With regard to the features that may arise in complex sets of PDEs or high-resolution data, the skip connections and nonlinear activation functions of the model architecture are much more capable of resolving such features. The Fourier layers serve to propagate information throughout the domain. These heuristic considerations have been backed by recent theoretical and empirical work in ArXiv:2210.01074 and ArXiv:2304.13221 and references therein.
> >
> > 5. We would like to address the reviewer's question on the **Full Grid baselines**. These "Full Grid" baselines use an integral kernel with the same number of modes as the SMM comparisons. Nonetheless, the FFT used in this baseline computes all the modes. The number of modes is indeed low enough for the sum to be directly evaluated, specifically through a PyTorch matrix multiplication function, and this is one of our primary contributions in this paper. This simple approach, as well as its extension to nonequispaced point clouds, is the key point of our paper and leads to a concrete realization of neural operators on arbitrary point clouds, a feature that was long promised in the literature but, to the best of our knowledge, is only provided in our paper.
> >
> > 6. We thank the reviewer for pointing out relevant references, and we have added them to the list of references in the revised version of our paper.
> >
> > We hope to have addressed the reviewer's concerns, particularly about the contributions of our paper vis-a-vis existing NUFFT libraries, to your satisfaction. We request the reviewer to kindly upgrade their assessment of our paper accordingly.

---

> > > ### Author Response · Authors · 2023-11-21
> > > **Requesting the reviewer for feedback**
> > >
> > > Due to the imminent closure of the discussion period, We kindly request the reviewer to provide us some feedback on our rebuttal. We are at your disposal for any further questions/clarifications regarding the paper or the rebuttal.

---

> ### Comment · Reviewer_7coN · 2023-11-22
> **Review updated**
>
> Since I am not sure if a notification gets sent out upon the update of a review, please note that I have updated my review after reading the authors response.

---

> ### Author Response · Authors · 2023-11-22
> **Reply to the reviewer's updated feedback**
>
> At the outset, we would like to sincerely thank the reviewer for their constructive feedback as well as for raising their rating for our paper. The reviewer is rightfully concerned about the **framing of our contributions** and we take this opportunity to further clarify our rationale for the choices that we had made in the original version of our paper. As we have stated in the paper and as the reviewer (and other reviewers) have acknowledged that our primary contribution is indeed the *realization of FNO (and related neural operators) on arbitrary point distributions*, a topic of great importance in the learning of partial differential equations. Clearly, our paper should not be construed as a contribution in the general area of *structured transforms* and we do apologize if our framing of the title, abstract and related work section led to a perception of contributions to structured transforms. We would like to rectify this misperception as much as possible in the **CRV**, if accepted. To this end, following the discussion with all the reviewers, we propose to change the title of the **CRV**, if accepted, to **Fourier Neural Operators on arbitrary point clouds** or **Neural Operators on arbitrary grids**. This will directly frame as contribution in terms of operator learning, rather than to structured transforms. Consequently, we will modify the abstract as well as the introduction, related work and discussion to very clearly state our exact contribution as we have framed it in our rebuttal to your evaluation. We will pay special attention to your comments on the related work section in this regard.
>
>  It is also pertinent here to point out that our choice of the title was motivated by the need to differentiate our paper from papers such as "Nonequispaced Fourier PDE Solvers" and "Geometry-Aware FNOs", which claim to realize FNO on arbitrary point clouds but do so via either interpolation or learnable transformations between irregular and regular grids. This is NOT what we do, so we chose a title that reflects this difference but we believe that the proposed modified title will reflect our true contributions in a much sharper manner.
>
> Finally, the reviewer rightly points out the use of FFT and Spherical harmonics as *red herrings*. In the original versions of FNO, the idea to use FFT was to reduce computational complexity as the authors thought that a large number of modes are necessary. This thinking has been persistent in the community ever since although many people have observed empirically that in practice, very low number of modes actually performs on par or better than large number of Fourier modes and the expressivity in FNO comes from large lifting layers, residual connections, and nonlinear activation functions. Theoretical understanding of this phenomena is much more recent, see ArXiv:2210.01074 and ArXiv:2304.13221. But it is precisely these observations that have motivated our approach which based on direct sum evaluation and leads to a very simple method for extending FNO to arbitrary point clouds, which is both more accurate as well as more computationally efficient than state of the art baselines. One could argue that this observation of ours was quite straightforward but to the best of our knowledge, we were first to do so and also to implement it at scale. This constitutes the core of our contributions in this paper and we aim to state it even more clearly and explicitly in a **CRV**, if accepted.
>
> We thank the reviewer again for your very constructive feedback that has helped us to better frame our contributions and significantly improve the quality of the paper. We hope to have addressed your concerns satisfactorily through this reply and if so, we kindly request you to upgrade your evaluation accordingly.

---

### Official Review · Reviewer_R5Ja · 2023-10-31

**Soundness:** 3 good
**Presentation:** 2 fair
**Contribution:** 3 good
**Rating:** 8
**Confidence:** 2

**Summary:**

The authors propose a method to widen the applicability of current efficient neural operator learning methods. Current Fourier neural operator learning method works efficiently by frequency domain transformation via Fourier transform. However, this requires that data is sampled at equispaced intervals. The authors propose a structured matrix method using Vandermonde-structured matrices, which can be used on data sampled at arbitrary locations. Through experiments, the authors show that the proposed method shows high accuracy in solving partial differential equations and can be trained efficiently.

**Strengths:**

The paper is very well written, the motivation is very clear. The experiments are comprehensive.

**Weaknesses:**

The authors could start their method discussion by introducing Fourier neural operator first, its mathematical formulation, and then pointing out where the structured matrix method is fitted in. At present, Figure-1 shows their workflow, however it's hard to visualize the change the proposed method is bringing about in the whole neural operator learning workflow.

**Questions:**

See weakness section.

---

> ### Author Response · Authors · 2023-11-17
> **Response to Reviewer R5Ja**
>
> We start by thanking the reviewer for your appreciation of the merits of our paper and your welcome suggestions to improve it. Below, we address the concerns raised by the reviewer and thank the reviewer in advance for their patience in reading our reply.
>
> 1. The reviewer's point about **introducing more information on FNO** is very well taken. Following your excellent suggestion, we have now added an entire section **SM** A.1, where we explicitly define the FNO. We have also referenced this section in the main text. However, given the space constraints, we could not incorporate a full section on FNO in the main text itself.
>
> 2. In response to the reviewer's point about **it being hard to visualize the proposed method is bringing about in the whole neural operator learning workflow in Figure 1**, we would like to say that we are primarily providing a more efficient method (SMM) that replaces the FFT and inverse FFT of the Fourier layer of the FNO and allows FNO to be realized on arbitrary point distributions. Other than that, the workflow is unchanged, and the FNO diagram would be identical. We have now explicitly mentioned this fact in the main text of the revised version of our paper.
>
> We hope that we have addressed the reviewer's concern to your satisfaction. If there are any further questions we could answer to upgrade the reviewer's confidence in their assessment of our paper, we would gladly do so.

---

### Official Review · Reviewer_6FVk · 2023-10-31

**Soundness:** 2 fair
**Presentation:** 3 good
**Contribution:** 3 good
**Rating:** 8
**Confidence:** 5

**Summary:**

The authors propose a generalisation of the Fourier and Spherical Fourier Neural Operators to arbitrary point clouds. The paper is fairly well-written, and generalises these methods to arbitrary grids via the utilisation of Vandermonde matrices and quadrature on unstructured grids. The authors apply this approach to a range of examples and provide good empirical evidence for the effectiveness of the method. I wish that the authors had stressed more that this is essentially the realisation of FNO/SFNO on arbitrary grids, something that the literature has promised but never delivered so far. As such, I believe that this paper is valuable to the community.

**Strengths:**

* The paper is well-written and well-motivated. The literature review is sufficient and mostly achieves to put the method into perspective (See my remarks in the Questions section)
* The authors address an interesting gap in the literature: the formulation of FNOs on arbitrary grids. To the best of my knowledge, this has been discussed in the literature, but has not been applied in practice.
* Ample benchmarks are provided

**Weaknesses:**

* The experimental results and the result tables would benefit from additional information (See my remarks)
* The text in the results section is a bit misleading and makes it sound as if the proposed method outperforms FNO and SFNO on the regular grid. If I understand correctly, this is the method on the irregular grid, which requires an interpolation step.
* It would have been better if the provided metrics were put into context by showing the approximation results of the vanilla FNO/SFNO methods on their respective grids. This would provide a good baseline both for timing and performance results.
* The given setting would allow to test operators tested on another grid to be evaluated on this unstructured grids. As I imagine this to be one of the main applications of this method, I would have like to see such results.

**Questions:**

* (Question) Classical methods typically can not perform arbitrarily well on unstructured grids. I expect the same to be true here as the projection will have quadrature errors and you won't be able to evaluate the integrals to high accuracy on arbitrary grids. Have you performed any analysis on how sensitive results are to the choice of collocation points? It would be good to mention this in the paper.
* (Remark) Related to the above question. it would be great to have both timings and L2/L1 errors of the FNO/SFNO on their respective grids to quantify the error of the method and put it into perspective. I expect these results to be better, but I would still welcome this to show the grid-dependence and avoid wrong conclusions for cases where the data is readily available on structured grids.
* (Remark) In both of the original FNO/SFNO papers, it is mentioned that the generalization to arbitrary grids is straightforward by formulating the DFT/SHT on the respective domain. I wish that the authors had stressed this aspect more in the theoretical introduction, as the presented method is essentially a realization of the FNO/SFNO, just on an unstructured grid. This is one of the significant advantages of FNOs, and it is great to see this done in practice.
* (Question) In line with what I wrote above, one of the advantages of Neural Operators is that they can be trained on one grid and evaluated on another. Have you experimented with such settings. I would be especially curious in the presented FNO/SFNO case how such an operator would perform compared to one trained on the unstructured grid

---

> ### Author Response · Authors · 2023-11-17
> **Response to Reviewer 6FVk**
>
> We start by thanking the reviewer for your appreciation of the merits of our paper and your welcome suggestions to improve it. Below, we address the concerns raised by the reviewer and thank the reviewer in advance for their patience in reading our detailed reply.
>
> 1. At the outset, we would like to thank the reviewer for pointing out that **our method is essentially the realization of FNO/SFNO on arbitrary grids, something that the literature has promised but never delivered so far.** We fully agree with your assessment in this regard. It is also true that we could have mentioned this point much more explicitly in the main text of our paper. One reason why we did not do so was to emphasize the *even broader scope* of our method. It is not limited just to FNO/SFNO, but in principle, any neural operator that uses spectral transformations (Fourier/Spherical Harmonics/Wavelet, etc.) can be efficiently realized on arbitrary point clouds using our structured matrix method. Already in the paper, we have demonstrated this with FNO, SFNO, UFNO, and FFNO (See Table 1), and the extension to other neural operators of this type is straightforward. However, we agree with the reviewer that FNO (and SFNO) are the most important neural operators of this class in practice. Hence, and following the excellent suggestion of the reviewer, we have explicitly (and repeatedly) pointed out in the revised version that our method realizes FNO/SFNO on arbitrary point clouds. We hope that this explicit message resonates much more with the end users of neural operators in scientific machine learning.
>
> 2. Regarding the reviewer's questions and comments about **how the proposed method compares to FNO/SFNO on regular grids and putting our method in the right perspective in this regard**: we would start by pointing out we had already provided such a comparison for the one-dimensional Burgers' Eqn (the very first two rows of Table 1) where we had presented the results for FNO on a uniform (equispaced) grid with the underlying Fourier transforms being computed by FFT and compared with the Fourier transforms being computed with SMM. From this table, we see that both methods are completely comparable in terms of training time as well as accuracy. Following your suggestion, we have performed an additional experiment for the 2-D shear layer with data sampled on a (dense) uniform grid for FNO, with FFT and SMM realizing the Fourier transform, respectively. We find that FNO with FFT has a per-epoch training time of $251$ secs with an $L^1$-test error of $6.76\%$. On the other hand, FNO with SMM has a per-epoch training time of $211$ sec and an $L^1$-test error of $6.6\%$. This clearly reinforces the conclusions of the 1-D experiment by demonstrating that, on a regular grid, SMM is completely comparable to FFT as they are simply different computational realizations of the Fourier transform inside FNO. We have now modified the main text of the paper to clearly point out this fact.
>
> Finally for the Shallow water equations on the sphere, we followed the reviewer's suggestion and trained SFNO (as well as FNO) on the underlying regular grid on a sphere. However, we evaluated both methods on randomly distributed points on the Sphere (outputs of SFNO and FNO can be evaluated on any point in the domain), and the resulting test errors are now shown in **SM Table 5**, as well as directly below. Comparing with the results obtained with SFNO-SMM (and FNO-SMM), we see that the errors are very similar whereas the SMM algorithm is faster to train in this case. With these additional results, we hope to have addressed your concerns about putting the results in the correct perspective by comparing with FNO/SFNO on regular grids.
>
> | Method | Training Time (per epoch) | Error over Collocation Points |
> |--------|---------------------------|-------------------------------|
> | SFNO   | 30s                       | 3.60%                         |
> | FNO    | 32s                       | 5.74%                         |
>
> 3. Regarding the reviewer's comments and questions about **how sensitive is the proposed method to the choice of collocation points**: the reviewer is correct in pointing out that, in analogy with classical numerical methods, one can expect larger errors on unstructured grids, in general. This is indeed borne out by the experimental evidence already presented in Table 1 for the 1-D Burgers' Eqn example. In the table, we present results on a regular equispaced grid and on an irregular expanding-contracting point distribution (See **SM** Figure 4) to observe that the test error with FNO-SMM increases by almost a factor of $4$ when transiting from a regular to an unstructured grid. We have added a further comment on this issue in the main text and thank the reviewer again for pointing it out to us.

---

> > ### Author Response · Authors · 2023-11-17
> > **Response to Reviewer 6FVk continued.**
> >
> > 4. In response to the reviewer's point about **advantages of FNO/SFNO is that they can be trained on one grid and evaluated on another**, we would like to say that this is indeed a very important feature of neural operators. As SMM is simply a realization of FNO/SFNO on arbitrary point clouds, the resulting method has exactly the same flexibility. Following the reviewer's excellent suggestion, we have now added a new Table (**SM** Table 5) to demonstrate this feature of the SMM in the specific case of Shallow water equations on a Sphere. To this end, we trained SFNO-SMM on a random point distribution of 5000 points and evaluated the trained model on random point distributions ranging from 4000 to 50000 points. The test errors with a test set of 64 samples are now shown in **SM** Table 5, and we see that the errors remain roughly the same even when the number of collocation points was increased by an order of magnitude. Moreover, we would like to emphasize that both the point distribution as well as the test samples were not seen during training. We believe that this example showcases the ability of the SMM algorithm to generalize to different arbitrary point clouds.
> >
> > | Approximate No. Points | Error  |
> > |------------------------|--------|
> > | 4000                   | 6.36%  |
> > | 8000                   | 5.64%  |
> > | 15000                  | 5.44%  |
> > | 29000                  | 5.39%  |
> > | 50000                  | 5.43%  |
> >
> > Furthermore, we would like to emphasize that in the Elasticity and Flow past Airfoil examples (Table 1), the point clouds changed for each sample during training (and testing) (See **SM** Figure 8). Thus, the SMM algorithm has the ability to deal with varying arbitrary point clouds.
> >
> > We sincerely hope that we have addressed all the reviewer's concerns, particularly about comparing to FNO/SFNO on regular grids and showcasing the ability of SMM to generalize to unseen point distributions, to your satisfaction. We kindly request the reviewer to update your assessment of our paper accordingly.

---

> > > ### Comment · Reviewer_6FVk · 2023-11-20
> > >
> > > I thank the authors for addressing my concerns and for adding additional experimental results to the manuscript. I am satisfied with the outcome and increased my rating to an accept.

---

> > > > ### Author Response · Authors · 2023-11-20
> > > > **Thanking the Reviewer**
> > > >
> > > > We sincerely thank the reviewer for your positive feedback on our rebuttal and also for increasing your rating. Your feedback has been very valuable in improving the quality of our paper.

---

### Author Response · Authors · 2023-11-17
**Response to All Reviewers**

At the outset, we would like to thank all three reviewers for their thorough and patient reading of our article. Their fair criticism and constructive suggestions have enabled us to improve the quality of our article. A revised version of the article is uploaded. We proceed to answer the points raised by each of the reviewers individually, below.

We would also like to point out that all the references to page numbers, sections, figures, tables, equation numbers and references, refer to those in the revised version.

Yours sincerely,

Authors of "A Structured Matrix Method for nonequispaced neural operators"

---

### Meta-Review · Area_Chair_bg3s · 2023-12-06

**Metareview:**

This paper extends neural operators like FNO to non-uniform grids, using an approach based on Vandermode-structured matrices. The authors provide an efficient PyTorch implementation and experimental evaluation on several benchmarks. While this paper does contain a clear contribution (an extension of neural operators and a solid implementation), the framing of the original submission was arguably misleading with regards to the contribution. In particular, the authors emphasized the algorithmic mechanisms used for the non-uniform grid case, though these techniques have been well-established in the non-neural operator literature. Moreover, there were some technical mistakes (e.g. conflating adjoint and inverse operators). The authors have already addressed many of these issues, including a reframing of the paper and a proposed new title. However, given that such a reframing is arguably a “major revision,” I believe this paper would benefit from an additional round of reviewing and should not be accepted as is. I hope that the authors are not discouraged by this decision, as I believe that this paper contains impactful contributions that will be best realized after the paper itself is revised.

**Justification For Why Not Higher Score:**

While the author's implementation and experimental results constitute a meaningful contribution, the paper itself requires a significant rewrite/reframing from the original submission. Thus, I believe this paper would benefit from an additional round of reviewing.

**Justification For Why Not Lower Score:**

N/A

---

### Decision · Program_Chairs · 2024-01-16

Reject